# Effect of intravitreal bevacizumab for retinopathy of prematurity on weight gain

**Shumpei Obata**[1], **Yusuke Ichiyama**[1]*, **Riko Matsumoto**[1], **Masashi Kakinoki**[1], **Yoshitsugu Saishin**[1], **Takahide Yanagi**[2], **Yoshihiro Maruo**[2], **Masahito Ohji**[1]

**1** Department of Ophthalmology, Shiga University of Medical Science, Otsu, Japan, **2** Department of Pediatrics, Shiga University of Medical Science, Otsu, Japan

* ichiyama@belle.shiga-med.ac.jp

## Abstract

### Purpose

To evaluate the short-term effect on body weight (BW) gain after intravitreal bevacizumab (IVB) for retinopathy of prematurity (ROP).

### Methods

This was a retrospective 1:1 matched case-control study. Infants with ROP treated by IVB or photocoagulation (PC) at Shiga University of Medical Science Hospital between April 2010 and December 2019 were included in the study. To match BWs at treatment between the IVB and PC groups, 1:1 matching for BWs at treatment within 100 g was performed. The BW gains for the 7 days before treatment (pre-treatment week), the 7 days after treatment (first post-treatment week), and the period from 7 to 14 days after treatment (second post-treatment week) were compared between the IVB and PC groups.

### Results

Following 1:1 matching, 13 infants in both groups were enrolled in the analysis. The weekly BW gain for the first post-treatment week was significantly lower in the IVB group compared with the PC group (86 g vs. 145 g; P = 0.046), whereas the weekly BW gains for the pre-treatment week (173 g vs. 159 g; P = 0.71) and the second post-treatment week (154 g vs. 152 g; P = 0.73) were comparable between the two groups. The short-term inhibitive effect of IVB on BW gain was particularly observed in infants weighing less than 1500 g at treatment (<1500 g: 47 g vs. ≥1500 g: 132 g; P = 0.03).

### Conclusion

IVB could have a short-term inhibitive effect on BW gain in infants with ROP, and this effect is more likely to occur in infants with a lower BW at the time of treatment.

**Data Availability Statement:** All relevant data are within the paper and its Supporting Information files.

**Funding:** M.O. received financial support only from Shiga University of Medical Science for this study. However, the funders had no role in study design, data collection and analysis, decision to publish, or preparation of the manuscript.

**Competing interests:** The authors have declared that no competing interests exist.

## Introduction

Retinopathy of prematurity (ROP) is a leading cause of childhood blindness [1]. Laser photocoagulation (PC) has been the gold-standard treatment for ROP, although it has ocular adverse effects, including laser-induced myopia and visual field loss [2–5]. In 2011, a multicenter randomized clinical trial demonstrated that intravitreal anti-vascular endothelial growth factor (VEGF) injection was effective for ROP treatment [6], and subsequent reports suggested that intravitreal anti-VEGF injection would have less ocular adverse effects, including myopia and visual field loss, compared with laser treatment [4,7,8]. On the basis of these studies, intravitreal anti-VEGF injection is becoming a first-line treatment for ROP instead of PC. However, recent clinical reports on the use of intravitreal anti-VEGF injections for ROP suggested that these anti-VEGF agents can subsequently escape from the eye into the systemic circulation [9–11], and thus potential adverse systemic effects after such injections should be considered.

In our previous study using a mouse model of ROP [12], a single intravitreal aflibercept injection inhibited body weight (BW) gain for one day post-injection. This result indicated that intravitreal anti-VEGF injection for ROP has a short-term adverse effect on BW gain. There are several reports on BW gain after intravitreal anti-VEGF injections for ROP, however, all of these reports focusing on BW gain after such injections analyzed only the long-term effect (from 2 to 28 months) [13–15]. Because long-term BW gain could be affected by various factors other than the anti-VEGF injection, including the amount of feeding and presence of diseases [16], it may be more appropriate to investigate short-term BW gain to examine the direct effect of anti-VEGF injection on BW gain. To the best of our knowledge, no report has evaluated the short-term effect on BW gain after intravitreal anti-VEGF injection, and this should be investigated.

The purpose of this study was to investigate the short-term effect of intravitreal anti-VEGF injection on BW gain compared with that of PC in infants with ROP.

## Methods

### Study design and patients

This retrospective 1:1 matched case-control study protocol was approved by the Institutional Review Board/Ethics Committee of Shiga University of Medical Science. All procedures involving human participants were performed in accordance with the ethical standards of the institutional and/or national research committee and with the 1964 Helsinki declaration and its later amendments or comparable ethical standards. For this study, an opt-out consent process was used at Shiga University of Medical Science Hospital following approval by the Institutional Review Board. The study included infants with type 1 ROP treated by intravitreal injection of bevacizumab (IVB) or PC at Shiga University of Medical Science Hospital between April 2010 and December 2019. Type 1 ROP was defined according to the Early Treatment for Retinopathy of Prematurity study [5]. The choice of whether to use PC or IVB was based on the parents' wishes after detailed explanation of the risks and benefits following detailed medical explanations by ROP specialists. We obtained written informed consent for treatment from the parents of each infant. Intravitreal injection of 0.625 mg bevacizumab (in 0.025 ml) was performed with a 29-gauge needle through the pars plana at 0.75–1.0 mm posterior to the limbus under general anesthesia, and off-label use of bevacizumab was approved by the Institutional Review Board of Shiga University of Medical Science. PC was also performed under general anesthesia.

To match BWs at the time of treatment between the IVB and PC groups, the following steps were performed: 1) infants without BW data at the time of treatment were excluded; 2)

infants in the IVB and PC groups were individually matched for BWs within 100 g; and 3) when there was more than one option for matching, the matching that minimized the difference between the two groups was performed.

## Data collection and outcomes

The following infant factors were investigated: sex, gestational age at birth, birth weight, Apgar scores at 1 and 5 minutes, multiple birth, respiratory distress syndrome, chronic lung disease, patent ductus arteriosus, intraventricular hemorrhage, necrotizing enterocolitis, periventricular leukomalacia, sepsis, and transfusion of red blood cells. Data for blood pressure, pulse rate, oxygen concentration set on the ventilator, volume of milk sucked, and urine output were collected from 7 days before to 14 days after the treatment. Data for BWs at treatment, 7 days before treatment, and 7 and 14 days after treatment were collected retrospectively from medical charts. In cases without BW data at these time points, predicted BWs were calculated from the two most recent points (e.g., for a case without BW data at 7 days after treatment, the median value of the BWs at 6 and 8 days after treatment was used to predict the weight at 7 days after treatment).

The BW gains for the 7 days before treatment (pre-treatment week), the 7 days after treatment (first post-treatment week), and the period from 7 to 14 days after treatment (second post-treatment week) were compared between the IVB and PC groups.

## Statistical analysis

Statistical analyses were performed using GraphPad Prism 8 software (GraphPad Software Inc., La Jolla, CA). Student's t-test or the Mann–Whitney U test was used for comparisons of continuous variables between the IVB and PC groups after the Shapiro–Wilk normality test. Fisher exact test was performed for comparisons of categorical data. P-values less than 0.05 were considered statistically significant.

## Results

Forty-six infants with type 1 ROP were treated at Shiga University of Medical Science Hospital during the study period. Thirty-four infants (IVB: n = 19; PC: n = 15) met the inclusion criteria for the study after excluding 12 infants because of a lack of data on their BW at the time of treatment. Following 1:1 matching for BWs at treatment within 100 g, 13 infants in each group were enrolled in the final analysis (Fig 1). All data are provided in the S1 File. The baseline characteristics are presented in Table 1. The numbers of boys (IVB: n = 9; PC: n = 3; P = 0.047) and zone 1 ROP cases (IVB: n = 6; PC: n = 1; P = 0.03) were significantly higher in the IVB group than in the PC group, whereas the other factors were comparable between the two groups, including gestational age, birth weight, postmenstrual age at treatment, and BW at treatment.

The mean BWs from 7 days before treatment to 14 days after treatment are shown in Fig 2A, and the weekly BW gains for the 7 days before treatment (pre-treatment week), the 7 days after treatment (first post-treatment week), and the period from 7 to 14 days after treatment (second post-treatment week) are shown in Fig 2B. The weekly BW gain in the IVB group for the first post-treatment week was significantly lower than that in the PC group (86 g vs. 145 g; P = 0.046), whereas the weekly BW gains for the pre-treatment week were comparable between the two groups (IVB: 173 g vs. PC: 159 g; P = 0.71). The post-treatment inhibitive effect of IVB on BW gain was no longer observed in the second post-treatment week (IVB: 154 g vs. PC: 152 g; P = 0.73). These results suggest that IVB has a short-term inhibitive effect on BW gain.

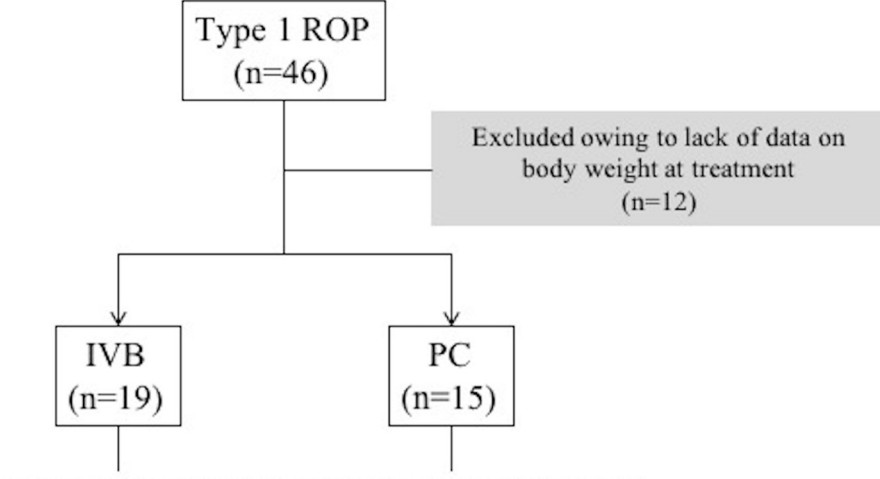

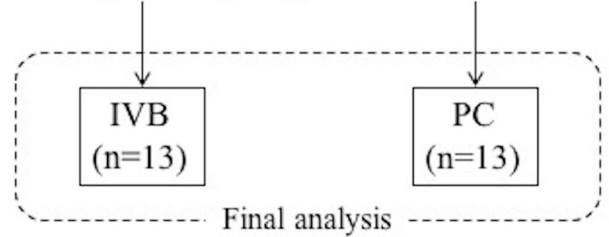

**Fig 1. Patient flow diagram.** ROP, retinopathy of prematurity; IVB, intravitreal injection of bevacizumab; PC, photocoagulation.

**Table 1. Baseline characteristics of the two groups after 1:1 matching for body weight at treatment within 100 g.**

| | IVB group n = 13 | PC group n = 13 | P-value |
|---|---|---|---|
| Sex (male/female) | 9/4 | 3/10 | 0.047 |
| Gestational age (week) | 25.8±2.3 | 26.5±1.6 | 0.10 |
| Birth weight (g) | 703±273 | 794±161 | 0.31 |
| Multiple birth | 4 | 2 | 0.64 |
| Apgar score at 1 minute | 4.4±2.3 | 4.2±2.3 | 0.80 |
| Apgar score at 5 minutes | 7.4±1.9 | 7.1±2.5 | 0.71 |
| Respiratory distress syndrome | 12 | 12 | >0.99 |
| Chronic lung disease | 8 | 9 | >0.99 |
| Patent ductus arteriosus | 11 | 11 | >0.99 |
| Intraventricular hemorrhage | 5 | 2 | 0.38 |
| Necrotizing enterocolitis | 2 | 0 | 0.48 |
| Periventricular leukomalacia | 0 | 1 | >0.99 |
| Sepsis | 1 | 2 | >0.99 |
| Transfusion of red blood cells | 9 | 6 | 0.43 |
| Postmenstrual age at treatment (week) | 36.2±2.6 | 36.3±2.4 | 0.89 |
| Body weight at treatment (g) | 1626±525 | 1621±542 | 0.98 |
| Days from birth to treatment (days) | 73±15 | 68±11 | 0.43 |
| Zone at treatment (I/II) | 7/6 | 1/12 | 0.01 |

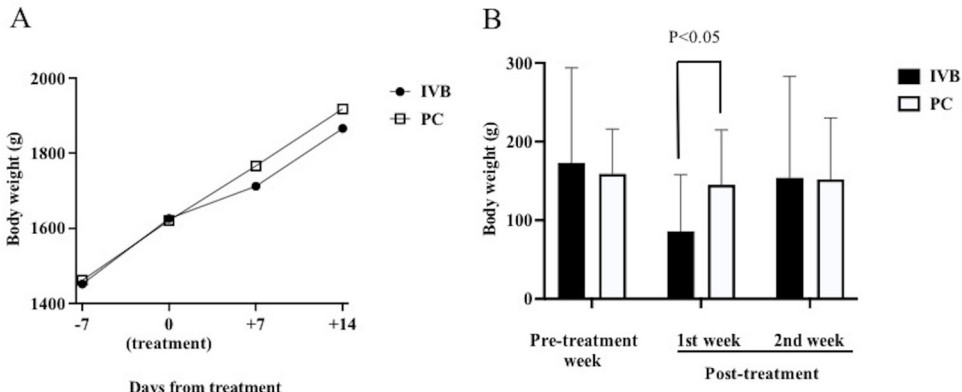

**Fig 2. BW pre- and post-treatment.** A, Mean BWs from 7 days before treatment to 14 days after treatment. B, Weekly BW gains for the 7 days before treatment (pre-treatment week), the 7 days after treatment (first post-treatment week), and the period from 7 to 14 days after treatment (second post-treatment week). BW gain in the IVB group during the first post-treatment week was significantly lower than that in the PC group. BW, body weight; IVB, intravitreal injection of bevacizumab; PC, photocoagulation.

To determine whether the effects of IVB on BW gain differed depending on the BW at treatment, we compared the weekly BW gains for the first post-treatment week between infants with BW <1500 g at treatment and infants with BW ≥1500 g (Fig 3). The weekly BW gains for the first post-treatment week in infants with BW <1500 g at treatment were significantly lower than infants with BW ≥1500 g at treatment in the IVB group (47 g vs. 132 g; $P = .03$) but not in the PC group (138 g vs. 158 g; $P = 0.73$).

The changes in blood pressure, pulse rate, oxygen concentration set on the ventilator, volume of milk sucked, and urine output from 7 days before treatment to 14 days after treatment are shown in Fig 4. There were no significant differences in these factors between the IVB and PC groups at each time point, although numerically there was an increase in oxygen

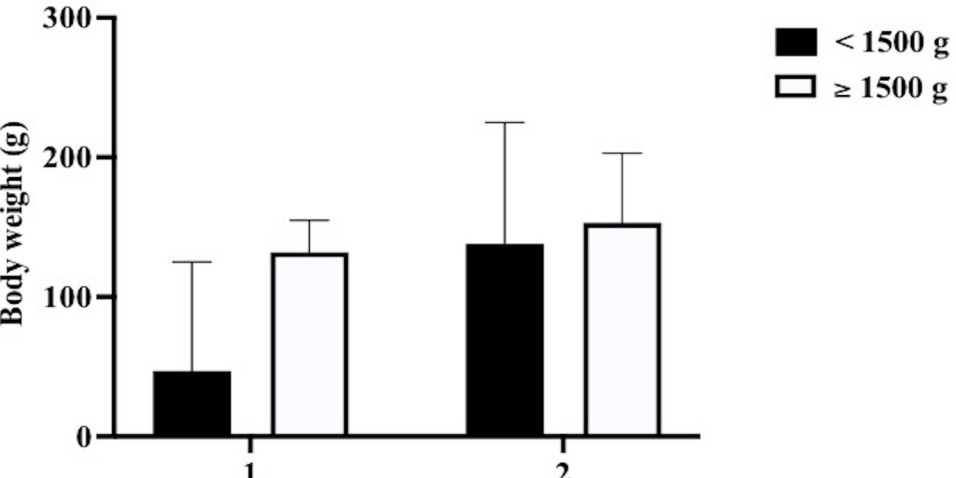

**Fig 3. Effect of Pre-treatment BW on BW gain post-treatment.** The BW gains during the first post-treatment week in infants with BW <1500 g at treatment and infants with BW ≥1500 g are shown. The BW gain in infants with BW <1500 g at treatment was significantly smaller than that in infants with BW ≥1500 g at treatment in the IVB group, but not in the PC group. BW, body weight; IVB, intravitreal injection of bevacizumab; PC, photocoagulation.

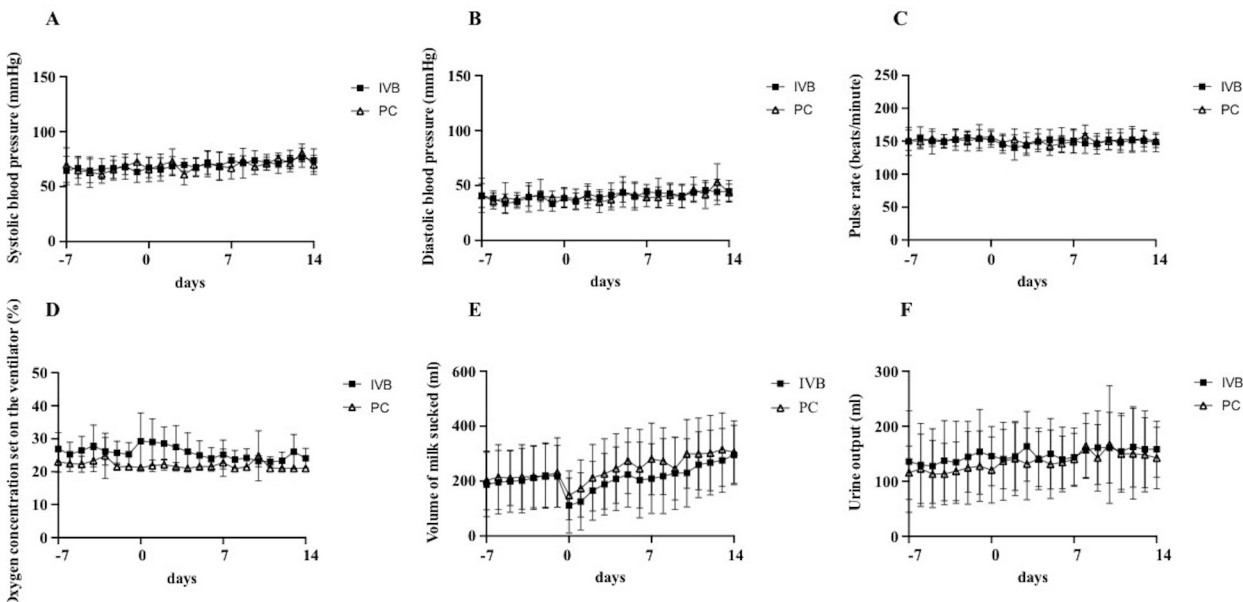

**Fig 4. Parameters other than BW gain.** A–F, Changes in systolic (A) and diastolic (B) blood pressure, pulse rate (C), oxygen concentration set on the ventilator (D), volume of milk sucked (E), and urine output (F) from 7 days before treatment to 14 days after treatment.

concentration set on the ventilator and a decrease in volume of milk sucked in the IVB group compared with the PC group.

## Discussion

We found that IVB displayed a short-term inhibitive effect on BW gain in infants with ROP, which is consistent with our previous research using a mouse model of ROP [12]. All of the previous clinical reports focusing on BW gain after intravitreal anti-VEGF injection analyzed only the long-term effect [13–15]. Because long-term BW gain could be affected by various factors other than anti-VEGF injection, including environmental factors, social factors, and various disorders [16], it may be more appropriate to investigate the short-term BW gain to determine the direct effect of anti-VEGF injection on BW gain. To the best of our knowledge, this is the first report demonstrating that intravitreal anti-VEGF injection has a short-term inhibitive effect on BW gain in human infants with ROP. We believe that the lower BW gain during the first post-treatment week in the IVB group compared with the PC group was a systemic adverse effect of IVB, because the BW gain during the pre-treatment week in the IVB group was similar to that in the PC group. In the present study, the ratios of boys and zone 1 ROP cases were significantly higher in the IVB group compared with the PC group. We do not believe that the difference in the ratio of boys would cause the difference in the BW gain in this study because BW gain was reported to be similar for boys and girls [17]. However, we cannot completely rule out the possibility that the different ratio of zone 1 ROP cases may be associated with the difference in post-treatment BW gain because no reports have compared BW gain between infants with zone 1 ROP and infants with zone 2 ROP. The required oxygen concentration before treatment was slightly higher in the IVB group, but the difference was not statistically significant. High oxygen concentrations interfere with the growth of retinal blood vessels, which may be related to the higher number of zone 1 ROP cases in the IVB group. Because there was no difference in BW gain before treatment between the two groups, whereas a difference in BW gain appeared after treatment between the two groups, it may be assumed

that the intervention of the treatment affected the results, rather than the difference in the zones as a baseline characteristic. We also found that the inhibitive effect of IVB on BW gain was more likely to occur in infants with low BW at treatment. A higher dose per BW of bevacizumab is administered to a smaller infant because the bevacizumab dose for ROP treatment is commonly fixed at 0.625 mg/0.025 ml [6]. Animal studies using mice also showed that increasing the anti-VEGF drug dose per BW suppressed BW gain to a greater extent [12,18]. Therefore, we should consider the use of lower doses of anti-VEGF agents, particularly in infants with smaller BW, because previous reports suggested that lower-than-standard doses were also effective for ROP [19,20].

Some of the previous reports focusing on systemic adverse effects of IVB suggested that it could cause neurodevelopmental disabilities in early childhood [15,21,22], while others showed no relationship between IVB and neurodevelopmental disorders [13,23,24]. Therefore, further analysis is needed to determine whether IVB causes neurodevelopmental disorders. The short-term inhibitive effect on BW gain after IVB found in the present study may be related to neurodevelopmental disorders.

To examine the difference in effects on parameters other than BW gain between the IVB and PC groups, we analyzed the changes in blood pressure, pulse rate, oxygen concentration set on the ventilator, volume of milk sucked, and urine output before and after treatment. The results revealed that the required oxygen concentration after treatment in the IVB group was higher than that in the PC group, albeit without statistical significance, suggesting that IVB treatment may have a negative effect on oxygenation in ROP infants. In addition, the volume of milk sucked was decreased in both groups after treatment, with a greater decrease in the IVB group compared with the PC group, again without statistical significance. Although it remains unclear how IVB causes a decrease in the volume of milk sucked, a decrease in the volume of milk sucked may be a reason for the short-term poor BW gain after IVB treatment.

The present study has several limitations. First, this was a retrospective study with a small sample size. Although we performed 1:1 matching for BW at treatment to reduce bias, a prospective study with a larger sample size is warranted in the future. Second, there were differences in baseline characteristics, including the ratios of boys and zone 1 ROP cases, which may cause lower BW gain; however, we believe that the smaller BW gain during the first post-treatment week in the IVB group compared with the PC group was a systemic adverse effect of IVB because the BW gains during the pre-treatment week were similar between the two groups. Finally, our conclusions cannot be applied to other anti-VEGF agents, including ranibizumab and aflibercept, because we only analyzed the effect of bevacizumab on BW gain. The effects of other anti-VEGF drugs on BW gain should be investigated in a future study.

## Conclusions

In summary, IVB could have a short-term inhibitive effect on BW gain in infants with ROP, and this effect is more likely to occur in infants with lower BW at the time of treatment. Because it remains unknown whether this short-term systemic adverse effect is associated with long-term disabilities, further investigation is warranted.

## Supporting information

**S1 File. Data set for analysis.**
(XLSX)

## Acknowledgments

We thank Shoji Momokawa and Jun Matsubayashi from Center for Clinical Research and Advanced Medicine, Shiga University of Medical Science, for their advice on statistical analyses. We thank Robert Blakytny, DPhil, and Alison Sherwin, PhD, from Edanz (https://jp.edanz.com/ac) for editing a draft of this manuscript.

## Author Contributions

**Conceptualization:** Yusuke Ichiyama.

**Data curation:** Shumpei Obata, Riko Matsumoto.

**Formal analysis:** Shumpei Obata.

**Funding acquisition:** Masahito Ohji.

**Investigation:** Shumpei Obata, Yusuke Ichiyama, Riko Matsumoto, Masashi Kakinoki, Yoshitsugu Saishin, Takahide Yanagi, Yoshihiro Maruo, Masahito Ohji.

**Methodology:** Shumpei Obata, Riko Matsumoto, Masashi Kakinoki, Yoshitsugu Saishin, Takahide Yanagi, Yoshihiro Maruo, Masahito Ohji.

**Supervision:** Yusuke Ichiyama, Yoshihiro Maruo, Masahito Ohji.

**Writing – original draft:** Shumpei Obata, Takahide Yanagi.

**Writing – review & editing:** Yusuke Ichiyama, Riko Matsumoto, Masashi Kakinoki, Yoshitsugu Saishin, Yoshihiro Maruo, Masahito Ohji.

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
