## [Decision Letter · Decision Letter 0]

23 Sep 2021

PONE-D-21-28331Effect of Intravitreal Bevacizumab for Retinopathy of Prematurity on Weight GainPLOS ONE

Dear Dr. Ichiyama,

Thank you for submitting your manuscript to PLOS ONE. After careful consideration, we feel that it has merit but does not fully meet PLOS ONE’s publication criteria as it currently stands. Therefore, we invite you to submit a revised version of the manuscript that addresses the points raised during the review process.

We look forward to receiving your revised manuscript.

Kind regards,

Vikas Khetan, MD

Academic Editor

PLOS ONE

Journal Requirements:

Additional Editor Comments:

A nice study.

If the authors can answer the queries raised by the reviewers, the article will look more enhanced.

Reviewers' comments:

Reviewer's Responses to Questions

**Comments to the Author**

1. Is the manuscript technically sound, and do the data support the conclusions?

Reviewer #1: Yes

Reviewer #2: Yes

2. Has the statistical analysis been performed appropriately and rigorously? 

Reviewer #1: Yes

Reviewer #2: Yes

3. Have the authors made all data underlying the findings in their manuscript fully available?

Reviewer #1: Yes

Reviewer #2: Yes

4. Is the manuscript presented in an intelligible fashion and written in standard English?

Reviewer #1: Yes

Reviewer #2: Yes

5. Review Comments to the Author

Reviewer #1: Well written article .Good research question .

It would also be interesting to have data regarding feeding habits during the period post bevacizumab

Was their a reduction in intake ?

Reactivation post bevacizumab is usually from 6 weeks to 35weeks as per Beat ROP study . Any specific reason for repeating intravitreal bevacizumab at 4 weeks

Reviewer #2: Since the basis of poor weight gain immediately after bevacizumab is attributed to some systemic effect which could be neurological and hormonal, were there any other side effects like changes in oxygen demands, urine output, blood pressure fluctuations, seizural activity or ever heart rhythm changes?

6. PLOS authors have the option to publish the peer review history of their article (what does this mean?). If published, this will include your full peer review and any attached files.

Reviewer #1: No

Reviewer #2: No

---

## [Author Response · Author response to Decision Letter 0]

4 Nov 2021

Point-by-Point Responses to Reviewers’ Comments

Re: “Effect of Intravitreal Bevacizumab for Retinopathy of Prematurity on Weight Gain” by Obata et al.

Please note that our revisions in the manuscript file are shown in red font.

Reviewer #1: Well written article. Good research question . It would also be interesting to have data regarding feeding habits during the period post bevacizumab

Was there a reduction in intake?

Response: Figure 4, page 6, lines 97–99; page 8, lines 148–158; page 11, lines 197–207. We thank the reviewer for these supportive comments. To examine the physiological changes before and after treatment, we additionally collected data on blood pressure, pulse rate, oxygen concentration set on the ventilator, volume of milk sucked, and urine output. The IVB and PC groups both showed a temporary decrease in the volume of milk sucked after treatment. The decrease was numerically greater in the IVB group compared with the PC group, but without statistical significance. Although we could not exclude the possibility that decreased feeding may have contributed to the poor BW gain, the mechanism by which IVB caused poor BW gain remained unclear.

Reactivation post bevacizumab is usually from 6 weeks to 35weeks as per Beat ROP study. Any specific reason for repeating intravitreal bevacizumab at 4 weeks

Response: At our institution, we do not re-administer bevacizumab at 4 weeks. 

 

Reviewer #2: Since the basis of poor weight gain immediately after bevacizumab is attributed to some systemic effect which could be neurological and hormonal, were there any other side effects like changes in oxygen demands, urine output, blood pressure fluctuations, seizural activity or ever heart rhythm changes?

Response: Figure 4; page 10, lines 175–179; page 11, lines 197–207. We thank the reviewer for the suggestions to improve the quality of our manuscript. To examine the physiological changes before and after treatment, we additionally collected data on blood pressure, pulse rate, oxygen concentration set on the ventilator, volume of milk sucked, and urine output as data that we could collect at our institution. As mentioned in the Discussion section, the required oxygen concentration before treatment was numerically slightly higher in the IVB group, but the difference was not statistically significant. High oxygen concentrations interfere with the growth of retinal blood vessels, which may be related to the higher number of Zone 1 cases in the IVB group. There were no changes in blood pressure, pulse rate, and urine output between before and after treatment in the IVB and PC groups. The required oxygen concentration after treatment in the IVB group was higher than that in the PC group, albeit without statistical significance, but it remained unclear whether the use of high oxygen concentrations had an effect on the poor weight gain and the mechanism for how high oxygen concentrations affected BW gain was also unclear. Both groups showed a temporary decrease in volume of milk sucked after treatment. The decrease was greater in the IVB group compared with the PC group, but without statistical significance. Although we could not exclude the possibility that decreased feeding may have contributed to the poor BW gain, the mechanism by which IVB caused poor BW gain remained unclear.

---

## [Decision Letter · Decision Letter 1]

24 Nov 2021

Effect of Intravitreal Bevacizumab for Retinopathy of Prematurity on Weight Gain

PONE-D-21-28331R1

Dear Dr. Ichiyama,

We’re pleased to inform you that your manuscript has been judged scientifically suitable for publication and will be formally accepted for publication once it meets all outstanding technical requirements.

Kind regards,

Vikas Khetan, MD

Academic Editor

PLOS ONE

Additional Editor Comments (optional):

Reviewers' comments:

Reviewer's Responses to Questions

**Comments to the Author**

1. If the authors have adequately addressed your comments raised in a previous round of review and you feel that this manuscript is now acceptable for publication, you may indicate that here to bypass the “Comments to the Author” section, enter your conflict of interest statement in the “Confidential to Editor” section, and submit your "Accept" recommendation.

Reviewer #1: All comments have been addressed

Reviewer #2: All comments have been addressed

2. Is the manuscript technically sound, and do the data support the conclusions?

Reviewer #1: Yes

Reviewer #2: Yes

3. Has the statistical analysis been performed appropriately and rigorously? 

Reviewer #1: Yes

Reviewer #2: Yes

4. Have the authors made all data underlying the findings in their manuscript fully available?

Reviewer #1: Yes

Reviewer #2: Yes

5. Is the manuscript presented in an intelligible fashion and written in standard English?

Reviewer #1: Yes

Reviewer #2: Yes

6. Review Comments to the Author

Reviewer #1: Supporting data has been collected to answer the comments. This paper provides insights into formulating further research question for future purposes.

Reviewer #2: (No Response)

7. PLOS authors have the option to publish the peer review history of their article (what does this mean?). If published, this will include your full peer review and any attached files.

Reviewer #1: **Yes: **Dr Suganeswari Ganesan

Reviewer #2: **Yes: **Chetan Rao

---

## [Editor Report · Acceptance letter]

3 Dec 2021

PONE-D-21-28331R1 

Effect of Intravitreal Bevacizumab for Retinopathy of Prematurity on Weight Gain 

Dear Dr. Ichiyama:

I'm pleased to inform you that your manuscript has been deemed suitable for publication in PLOS ONE. Congratulations! Your manuscript is now with our production department. 

Kind regards, 

on behalf of

Dr. Vikas Khetan 

Academic Editor

PLOS ONE